# Sensitivity Analysis of Structural Parameters of Unequal-Span Continuous Rigid Frame Bridge with Corrugated Steel Webs

**Maojun Duan *** , **Fei Wang, Yutian Wu, Hao Tao and Danping Zhang**

College of Civil Engineering, Nanjing Forestry University, Nanjing 210037, China; wanafei@njfu.edu.cn (F.W.); wuyutian@njfu.edu.cn (Y.W.); 17805246869@njfu.edu.cn (H.T.); zhangdanping@njfu.edu.cn (D.Z.)

\* Correspondence: dmj@njfu.edu.cn

**Abstract:** To investigate the effect of structural parameters of bridges with unequal spans on the bridge alignment, the finite element model simulating the full-scale bridge was developed, considering the construction process. For ease of finite element modeling and investigation, the section of composite beam with corrugated steel web was first converted into the section composed of the same material. For this purpose, an equivalent method of replacing corrugated steel webs with concrete webs was proposed based on theoretical derivation. After equivalent replacement, the influences of material bulk density, internal prestress, pipe friction coefficient, and pipe deviation coefficient on the main beam at the maximum cantilever stage were analyzed, and the influences of external prestress on the main beam after bridge construction were analyzed. The results show that the most sensitive parameter to structural response is bulk density, subsequently the external prestress, internal prestress, pipe friction coefficient, and pipe deviation coefficient. Among them, the bulk density, internal prestress, and external prestress are all sensitive parameters, while pipe friction coefficient and pipe deviation coefficient are non-sensitive parameters.

**Keywords:** structural parameters; equivalent substitution; sensitivity analysis; unequal span

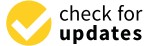



## 1. Introduction

The cantilever construction of a long-span continuous rigid frame bridge is affected by many factors, such as the structural bulk weight, construction load, loss of internal and external prestress, etc., which makes the construction control of long-span continuous rigid frame bridges much more complicated [1,2]. In the process of bridge design, due to the unknown real value of construction parameters, a reasonable value is generally assumed. Subsequently, the upper and lower edge stress and deflection changes in the structure in each construction stage could be determined through analysis and calculation. In this case, the state of the bridge could be in line with the theoretical state [3–5]. However, in the actual construction process, some parameters will change due to the influence of various factors, which leads to the deviation of the construction state from the design, and may eventually lead to a certain deviation between the linear shape of the bridge and the theoretical situation. Therefore, the primary problem of construction control is to adjust the parameters to ensure that they are closer to the actual value. In this case, the stress state and geometric linear of the bridge could remain as close to the theoretical values as possible. Therefore, it is necessary to carry out sensitivity analysis on these parameters, confirm the sensitive and insensitive parameters affecting the linear shape of the main beam, and control them in the construction process [6].

Extensive investigations regarding the sensitivity of the structural parameters of bridges have been carried out. Wang et al. [7] analyzed the sensitivity of parameters such as the bulk density, non-uniform temperature change, shrinkage, and creep of the main bridge of Longhua Songhuajiang River by the finite element method. Zhao et al. [8] analyzed the sensitivity of material bulk density, prestress, and other parameters to mechanical

properties of the maximum cantilever stage in combination with practical engineering. Wang et al. [9] investigated the influence of concrete density, temperature change, and other parameters on bridges based on practical engineering. Yang et al. [10] investigated the sensitivity of corrugated steel web bridges. Jiang et al. [11] analyzed the sensitivity of various parameters based on uniform design and regression analysis. Xu et al. [12] analyzed the sensitivity of some parameters of cable-stayed bridges. Mei et al. [13] proposed a new method for sensitivity analysis to balance computational efficiency and structural parameter changes, which introduced a parameter substructure to determine random variables. Chen et al. [14] proposed a sensitivity index calculation method based on an artificial neural network and performed the application to cable-stayed bridges. Pan et al. [15] proposed a parameter sensitivity analysis based on neural network integration and then applied it to prestressed concrete bridges. Zheng et al. [16] carried out the parameter sensitivity analysis of long-term deflection by using a numerical simulation method, which was helpful to reveal the main mechanism of long-term deflection of continuous rigid frame bridges. Cheng et al. [17] conducted a sensitivity analysis on the design parameters of a long-span mixed-beam cable-stayed bridge to improve the construction control precision. Choi et al. [18] investigated the sensitivity of structural parameters involved in the construction of multi-span continuous girder bridges with corrugated steel webs to improve monitoring accuracy. Overall, sensitivity analysis of structural parameters has been carried out as mentioned above; however, few studies focus on the effects of structural parameters on unequal span bridges, which warrants further research.

Based on practical engineering, this research focused on the investigation of influences of bulk density of the main beam, internal prestress, pipe friction coefficient, and pipe deviation coefficient on a deflection at the maximum cantilever stage, and the investigation of the influence of internal prestress on a deflection at the state of bridge completion, and determining the sensitivity of aforementioned parameters to the bridge structure. For this purpose, a simple variable method (i.e., assuming that there was no influence among structural parameters, and then only changing the value of a single variable) was adopted, and the results were obtained from the finite element modeling and analyzed.

## 2. Background of Project

The Nanjing Xinshengwei Yangtze River Bridge was selected as the object of the present study. The bridge is a double-tower single-span suspension bridge with a main span of 1760 m. The first link of the north approach bridge is an unequal-span continuous rigid frame bridge with corrugated steel web and prestressed concrete. The north approach bridge is 5 spans with a span of (50 + 60 + 80 + 100 + 70) m and a total length of 360 m. The web of the bridge uses a corrugated steel web with a height of 3.36 m. The roof and floor adopt C55 concrete, and the concrete density is 25 kN/m$^3$. Q345qD-typed steel with a bulk density of 78.5 kN/m$^3$ is used in the bridge construction. Both external and internal prestressed wires are employed, and the prestressed tension control stress is 1860 MPa. The cross-section of the first span is equal, while the one of the second to the fifth span is variable. The bottom shape of the beam changes according to the circular curve. The overall layout and real view of the north approach bridge are shown in Figures 1 and 2.

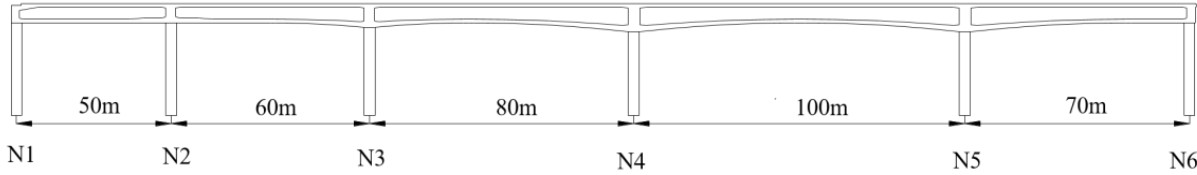

**Figure 1.** General arrangement of the north approach bridge of Nanjing Xinshengwei Yangtze River Bridge, where N1 represents the main tower, while N2, N3, N4, N5, and N6 represent the piers of the north approach bridge.

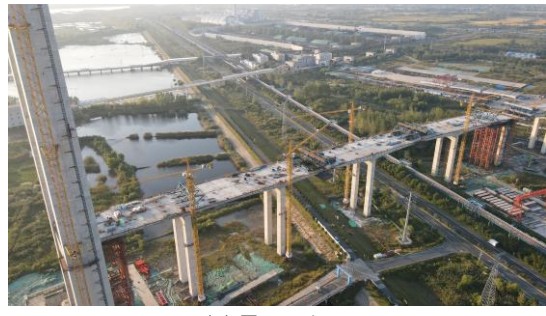

(**a**) Top view

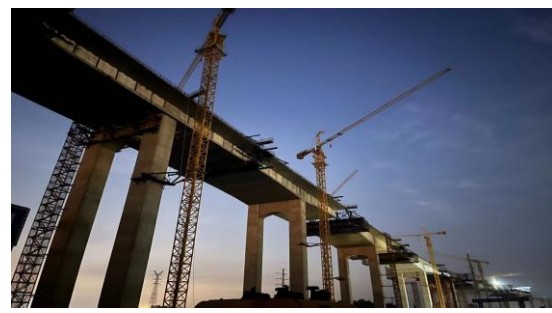

(**b**) Side view

**Figure 2.** Actual scene of the north approach bridge.

### 3. Equivalent Conversion of the Cross-Section

Several investigations focused on bridges with corrugated steel webs. Tuo et al. [19] investigated the buckling strength of corrugated steel webs. Ren et al. [20] investigated the flexural performance of composite girder bridges with corrugated steel webs. Nie et al. [21] conducted an experimental study on the shear performance of bridges with corrugated steel webs. In these investigations, finite element analysis of the bridges simulated by solid elements was performed; however, the whole calculating process is not very efficient. Instead, relevant researchers proposed some equivalent calculation methods for bridges with corrugated steel webs. Ji et al. [22] simplified three-dimensional corrugated steel webs into two-dimensional models through the methods of stiffness equivalence and displacement equivalence. Mo et al. [23] simplified the three-dimensional corrugated steel web into a two-dimensional model through the longitudinal equivalent stiffness. Ma et al. [24] simplified three-dimensional corrugated steel webs into two-dimensional webs through stiffness equivalence and carried out static analysis. Zhou et al. [25] simplified the three-dimensional corrugated steel web into concrete web by integrating the theory of equal weight and equal stiffness. In general, in the current research, a method for replacing the corrugated steel web with equivalent concrete was proposed. Subsequently, the sensitivity of structural parameters in the construction stage was analyzed.

The beam model simplifies the three-dimensional structure into a one-dimensional model, and the deflection equation of the beam can be obtained by assuming that the fibers on each section of the beam remain in a straight line when bending and deforming. In practical applications, the boundary conditions can be determined according to the specific conditions, such as beam support conditions and boundary fixed conditions, and then the equation can be solved by a numerical solution or analytical solution to obtain the deflection distribution of the corrugated steel web under the action of external loads [26,27]. It should be noted that the above modeling process is based on the assumptions of Euler–Bernoulli beam theory. If the effects of shear deformation and moment of inertia need to be considered, the Timoshenko beam theory or Reddy beam theory can be used for analysis and modeling [28,29].

The cross-section of the composite beam with corrugated steel web is composed of steel and concrete. To improve the efficiency of finite element modeling without reducing the accuracy, it is necessary to convert the composite beam with the corrugated steel web into a section composed of the same material.

The basic principle of the conversion process is to keep four parameters unchanged, i.e., the resultant force before and after the conversion, the height of the centroid (i.e., the position of the resultant force), the overall dead weight of the box girder, and the strain coordination condition.

The labels $E_c$, $A_c$, $\varepsilon_c$, and $\sigma_c$ represent the elastic modulus of concrete, the cross-section area, strain, and stress, respectively. The labels $E_s$, $A_s$, $\varepsilon_s$, and $\sigma_s$ represent the elastic modulus of steel, the cross-section area, strain, and stress, respectively. $A_{sc}$ represents the area of the concrete section converted from the steel section.

According to the conversion conditions of cross-section, Equation (1) could be derived based on the unchanged resultant force:

$$\sigma_s A_s = \sigma_c A_{sc} \tag{1}$$

Equation (2) could be derived based on the unchanged strain coordination condition:

$$\frac{\sigma_s}{E_s} = \frac{\sigma_c}{E_c} \tag{2}$$

The elastic modulus ratio ($\alpha_{Es}$) of steel and concrete materials could be expressed by Equation (3):

$$\alpha_{Es} = \frac{E_s}{E_c} \tag{3}$$

Based on Equations (1)–(3), $A_{sc}$ could be expressed by either one of Equations (4)–(6):

$$A_{sc} = \frac{\sigma_s}{\sigma_c} A_s \tag{4}$$

$$A_{sc} = \frac{E_s \varepsilon_s}{E_c \varepsilon_c} A_s \tag{5}$$

$$A_{sc} = \alpha_{Es} A_s \tag{6}$$

Equation (7) could be given based on the unchanged overall dead weight of box girder before and after conversion:

$$g_s = g_c = \gamma_s A_s l \tag{7}$$

where $g_s$ represents the dead weight of steel web in composite beam, and $g_c$ represents the dead weight of the concrete web converted from $g_s$. In addition, $l$ is the length of the box girder length.

On this basis, $A_{sc}$ could be further expressed by Equation (8):

$$A_{sc} = \frac{2g_s}{\gamma_c l} \tag{8}$$

Assuming that $t_s$ represents the thickness of the corrugated steel web plate section, and $t_c$ represents the thickness of the converted concrete web. Based on the fact that the height of box girder cross-section remains unchanged, the thickness of the concrete web after conversion could be expressed by Equations (9) and (10):

$$t_c = \frac{2g_s}{\gamma_c l h} = \frac{A_{sc}}{h} = \frac{\alpha_{Es} A_s}{h} \tag{9}$$

$$t_c = \alpha_{Es} t_s \tag{10}$$

where $h$ is the height of the box girder.

In the modeling process, it is assumed that the corrugated steel web is completely connected with the concrete roof and bottom plate, and the slip between the corrugated steel web and the top and bottom plate is ignored. Assuming that the two parts are completely bonded, and then using the engraving function, the curve of the corrugated steel web is engraved on the concrete top and bottom plate, so that the corrugated steel web and the concrete top and bottom plate can be jointed. To verify the accuracy of the equivalent conversion method, a N4 cantilever segment was taken as an example. The bulk density of steel is about three times that of concrete. The elastic modulus of concrete is $3.55 \times 10^4$ MPa and that of steel is $2.09 \times 10^5$ MPa. The elastic modulus of steel is about 5.8 times that of concrete. The solid element model simulating the N4 cantilever segment is shown in Figure 3, and the linkage model after section conversion is shown in Figure 4.

According to the derived Equation (10), a 127.6 mm thick concrete web was modeled in the linkage model instead of the 22 mm thick corrugated steel web simulated in the solid model.

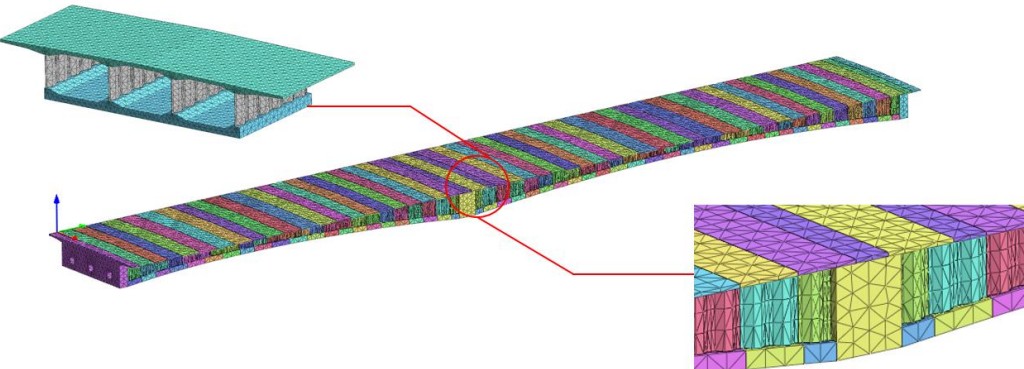

**Figure 3.** Solid model simulating the N4 cantilever segment.

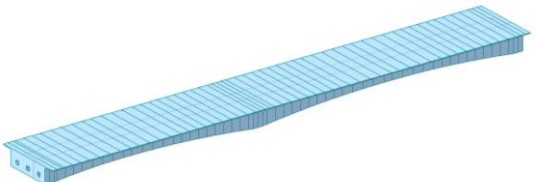

**Figure 4.** Linkage model simulating the N4 cantilever segment.

Only the dead load was applied to the finite element models. The calculation results of the two models are extracted to obtain the longitudinal variation diagram of deflection along the whole bridge, as plotted in Figure 5.

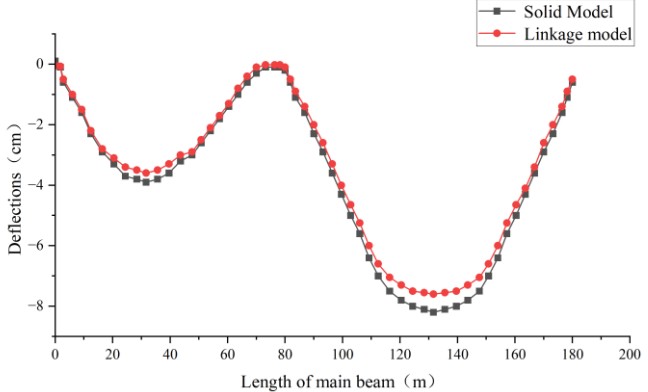

**Figure 5.** Comparison of the results obtained from solid and linkage models.

As can be seen from Figure 5, after equivalent conversion of the cross-section with corrugated steel web, the deflection curve of the linkage model has the same distribution trend as that of the solid model. The maximum deflection occurs at the mid-span position, and the difference between the frame and solid models is greatest at the mid-span position. The maximum deflection at the mid-span position was 7.2 mm for the linkage model and 8.2 mm for the solid model, with a maximum deflection difference of 0.6 mm. The deflection of the solid model was 7% greater than that of the linkage model. Zhang [30] investigated the mechanical characteristics of composite box beams with corrugated steel webs by using a simplified method integrating equal stiffness and equal weight, which is similar to the proposed equivalent conversion method in this study. Results show that the maximum error of deflection under the action of self-weight is 7%, which is the same as the

conclusion obtained by the method proposed in this paper. Therefore, it is believed that the equivalent conversion method proposed in the present study is feasible and the obtained results are acceptable.

In summary, after equivalent conversion, the deflection curves of the linkage model and solid model have almost the same distribution trend, and the maximum deflection difference is small, which verifies the feasibility and reliability of the equivalent conversion method. Therefore, the equivalent method can be adopted and a simpler model simulated by the linkage element can be used for analysis. By using the equivalent model, the calculation amount of composite concrete beams with corrugated steel webs can be reduced to some extent, the calculation can be simplified, and the efficiency can be improved. In practical engineering, the complex three-dimensional structure can be simplified into a one-dimensional model, which can greatly simplify the analysis process and reduce the amount of calculation and time, and the simplified model is usually based on relatively simple assumptions and theories, which is easier to understand and apply to practical engineering problems. However, the accuracy of a simplified model is slightly lower than that of a solid model, and structural details are often ignored in simplified models.

## 4. Structural Sensitivity Analysis

The developed finite element model simulating the three-box corrugated steel web bridge was shown in Figure 6. According to the characteristics of the bridge, five random variables are selected as the sensitivity analysis indexes, namely, the volume weight of the main beam, the internal prestress, the external prestress, the friction coefficient of the pipeline, and the deviation coefficient of the pipeline. The variation ranges of the above five random variables were determined, which were −5%, −10%, 5%, and 10%, respectively. The deflection of the main beam was selected as the sensitivity analysis index. The changed values of the parameter were substituted into the finite element model and calculated. The maximum deflection at the cantilever end and the maximum deflection of each span were extracted, and the sensitivity of the selected parameters was sorted.

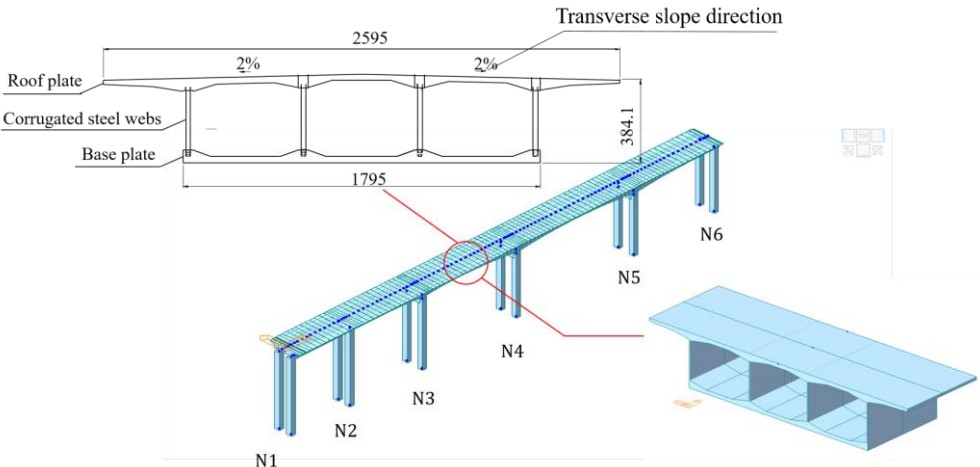

**Figure 6.** Full-scale finite element model.

### 4.1. Sensitivity Analysis of the Maximum Deflection to the Bulk Density of the Main Beam

The concrete is composed of coarse aggregate, sand, water, etc. in a certain proportion of mixing. The mixture ratio is determined in the design stage, but the amount of material and the specified value is difficult to be the same in the construction process, which will lead to changes in the concrete bulk density. For consideration of structural safety, it is necessary to analyze the sensitivity of the bulk density of the main beam.

As plotted in Figure 7, at the state of the maximum cantilever, as the increased bulk weight of the main beam leads to the increasing dead weight and the original prestressed reinforcing bars are insufficient, the closing opening of each span will bend down with the

increasing bulk weight, and the trend of bending down is linear. The relationship between the deflection and bulk density of each span is different. The deflection at the second span is smaller than that at the other four spans. With the variation of the bulk density, the variation of the maximum deflection at the mid-span position of each span increases (or decreases) with the increasing (or decreasing) bulk density, as shown in Figure 8. Under the condition of the maximum cantilever stage, the variation of maximum deflection at the mid-span position of each span varies linearly with the bulk density. Comparatively, the deflections at the first and second spans are less sensitive to the variation of the bulk density than the other three spans. The deflection at the fifth span is the most sensitive, i.e., the deflection changes 26% for every 5% change in bulk density. Above all, the change in bulk density has a great influence on the linear shape of the bridge, which will lead to a great deviation between the linear shape of the completed bridge and the theoretical situation. Therefore, the sensitivity analysis of bulk density provides guidance and a decision-making basis, which is helpful to optimize the design, improve the engineering quality, and ensure the safety of the project.

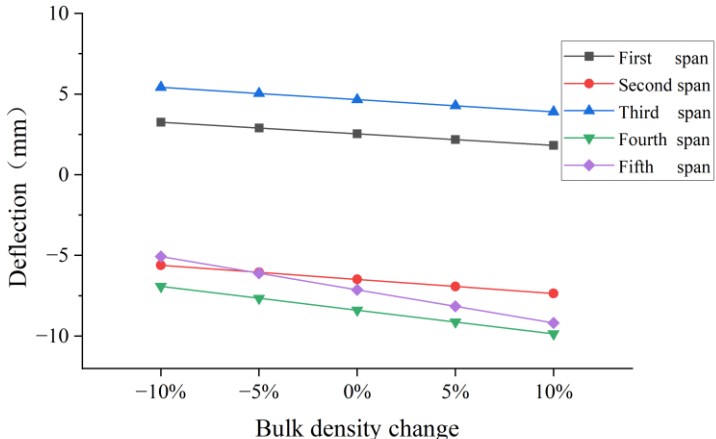

**Figure 7.** Variation of the deflection per span at maximum cantilever end.

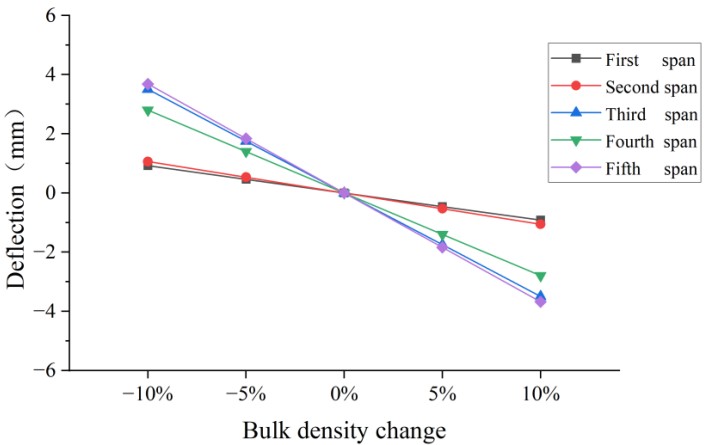

**Figure 8.** Relationship between the bulk density and maximum variation value of deflection per span.

### 4.2. Sensitivity Analysis of the Maximum Deflection to the Internal Prestress

When anchoring the steel bundle, the actual internal prestress during the construction process is different from the designed value due to non-standard operation, which will lead to an inevitable impact on the linear shape of the main beam. Therefore, sensitivity analysis of the internal prestress is conducted in this section.

As plotted in Figure 9, it can be seen that at the state of the maximum cantilever, the effect of the internal prestress increases with the increasing value. Therefore, the end of

the closure segment of each span is tickled upward with the increasing internal prestress, and the upward torsion trend is linearly varied. After changing the internal prestress, the maximum variation of the deflection of each span occurs at the mid-span position, as plotted in Figure 10. Under the maximum cantilever condition, the variation of maximum deflection at the mid-span position of each span presents a linear relationship with the variation of the internal prestress. The deflections of the first and second spans are less sensitive to the variation of internal prestress compared to the other three spans. Among them, the deflection of the fourth span is most sensitive to the variation of internal stress. When the internal prestress changes by 5%, the deflection changes by 5%. Above all, the change in internal prestress has a certain influence on the linear shape of the bridge, which reflects the problems of structural stability and design optimization. These issues are of great significance to the safety and sustainability of the structure, and by analyzing the data laws, the prestressed design can be optimized and the performance and service life of the structure can be improved.

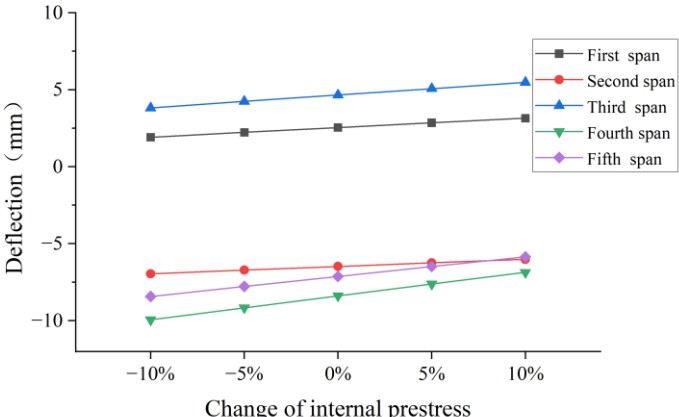

**Figure 9.** Variation of the maximum deflection at the cantilever end per span.

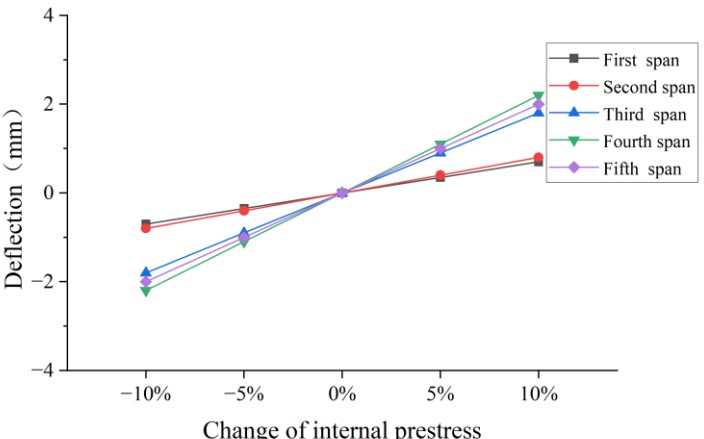

**Figure 10.** Relationship between the change in internal prestress and the maximum variation value of the deflection per span.

### 4.3. Sensitivity Analysis of the Maximum Deflection to the Pipeline Friction Coefficient

The pipe friction coefficient is determined by the material itself, so it will inevitably lead to the change in tension control stress, which directly affects the change in structural deflection [31,32]. Therefore, it is necessary to analyze the sensitivity of the pipeline friction coefficient, and subsequently control the value of the pipe friction coefficient.

As shown in Figure 11, under the condition of the maximum cantilever, the maximum deflection at the cantilever end per span is insensitive to the variation of the pipeline friction coefficient. After changing the pipe friction coefficient, the maximum variation of

the deflection of each span occurs at the mid-span position, as shown in Figure 12. Under the condition of the maximum cantilever, the maximum variation value of the deflection at the mid-span position of each span presents a linear relationship with the change in pipe friction coefficient. The variation ranges of the deflection corresponding to the first and second spans are the same and less than that of the other three spans. The variation values of the deflection of the third and the fifth span are the same. Among them, the deflection of the fourth span is most sensitive to the variation of the pipe friction coefficient. When the pipe friction coefficient changes by 5%, the deflection changes by 0.3%. Above all, the change in pipeline friction coefficient has almost no effect on the bridge alignment, which is a non-sensitive parameter.

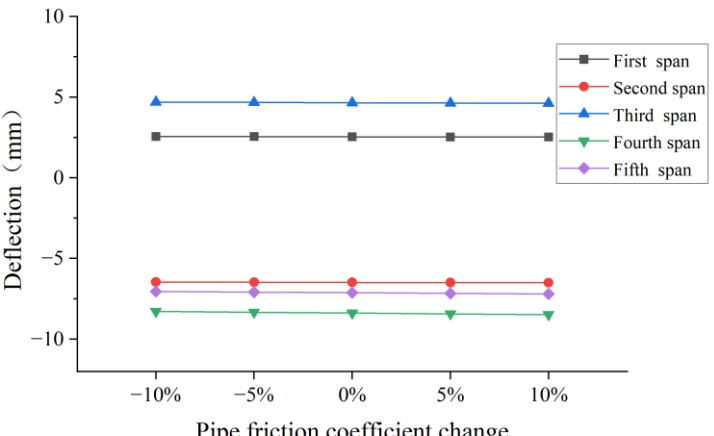

**Figure 11.** Variation of the maximum deflection at the cantilever end per span.

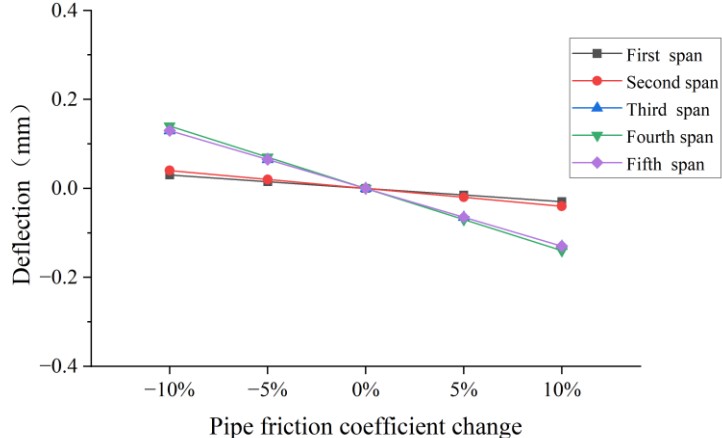

**Figure 12.** Relationship between the change in pipe friction coefficient and the maximum variation value of the deflection per span.

### 4.4. Sensitivity Analysis of the Maximum Deflection to the Pipeline Deviation Coefficient

The pipeline deviation coefficient is determined by the material itself, so it will inevitably lead to the change in tension control stress, which directly affects the change in structural deflection. Therefore, the control analysis of the pipeline deviation coefficient is essential in the monitoring.

As shown in Figure 13, under the condition of the maximum cantilever, the maximum deflection at the cantilever end remains unchanged with the variation of the pipeline deviation coefficient. After changing the pipeline deviation coefficient, the maximum variation value of the deflection of each span occurs at the mid-span position, as shown in Figure 14. Under the condition of the maximum cantilever, the maximum variation value of the deflection at the mid-span position of each span varies linearly with the variation of the

pipeline deviation coefficient. The variation ranges of the deflection corresponding to the first and second spans are smaller than that corresponding to the other three spans. Among them, the deflection corresponding to the fourth span is most sensitive to the variation of the pipeline deviation coefficient. When the pipeline deviation coefficient changes by 5%, the deflection changes by 0.2%. Above all, the variation of the pipeline deviation coefficient has almost no impact on the bridge alignment, which is also a non-sensitive parameter.

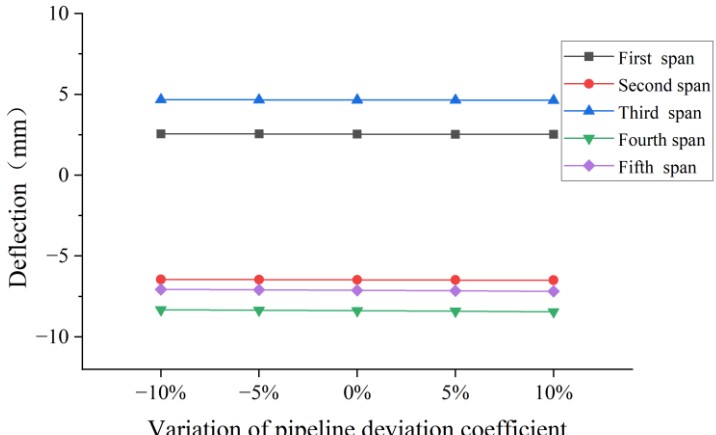

**Figure 13.** Variation of the maximum deflection at the cantilever end per span.

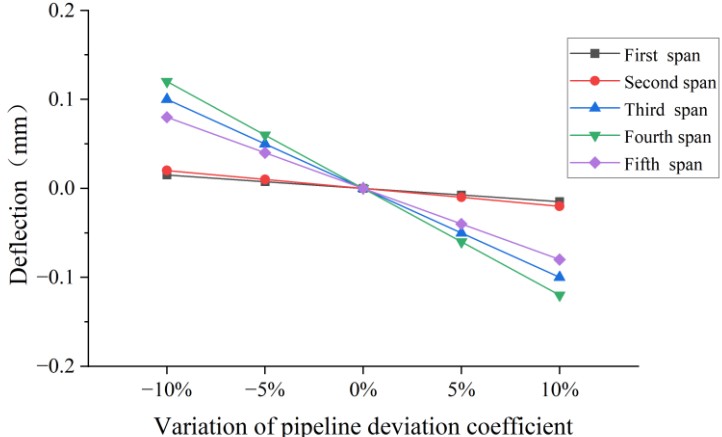

**Figure 14.** Relationship between the pipeline deviation coefficient and the maximum variation value of the deflection per span.

### 4.5. Sensitivity Analysis of the Maximum Deflection to the External Prestress

Since the external prestressed reinforcing bars are stretched after bridge closure, the sensitivity analysis of external prestressed steel is carried out after the construction process. The small amount of the external prestressed anchorage member can reduce the dead weight of the bridge and greatly improve the structural bearing capacity. Because the external prestress is arranged outside the concrete section and easily subject to corrosion, it is vulnerable to reducing the efficiency of the external prestress, which will in turn have an impact on the deformation and stress of the whole bridge.

As shown in Figure 15, after the bridge closure, the maximum deflection at the cantilever end increases with the increasing external prestress. The variation trend is linear; however, the variation range is small. After changing the external prestress, the maximum variation value of the deflection at each span occurs in the mid-span position, as plotted in Figure 16. With the variation of the external prestresses, the maximum variation value of the deflection at the mid-span position of each span presents a linear variation trend. In the case of the same variation range of the external prestress, the maximum

variation value of the deflection corresponding to each span is linear. The deflection corresponding to the fifth span is most sensitive to the variation of the external stress. When the external prestress changes by 5%, the deflection changes by 6%. Above all, the change in external prestress has a certain influence on the linear shape of the bridge, which reflects the problems of structural stability and construction quality control. These problems are of great significance to the safety, stability and sustainability of the structure. By analyzing the data law, the prestressed design can be optimized and the stability and durability of the structure can be improved.

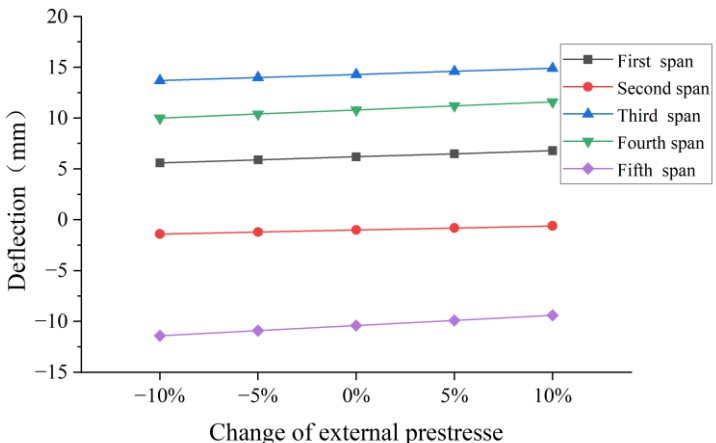

**Figure 15.** Variation of the maximum deflection at the cantilever end per span.

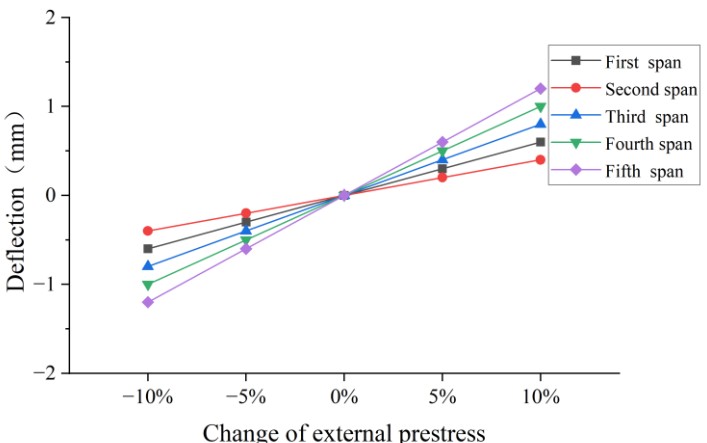

**Figure 16.** Relationship between the external prestress and the maximum variation value of the deflection per span.

### 4.6. Sensitivity Analysis of the Maximum Deflection to the Boundary Conditions

The abovementioned analysis was based on the specific boundary conditions which was in consistent with the real bridges. Hence, it is necessary to clarify whether the obtained results are valid for various boundary conditions. For this purpose, three types of boundary conditions were considered to investigate the relationship between the bulk density and maximum variation value of deflection per span: (1) the hinged connection was modelled between the pier labelled N3 and the main beam, while the rigid connection was considered between the other piers and the main beam (labelled boundary condition A); (2) the hinged connection was modelled between the pier labelled N4 and the main beam, while the rigid connection was considered between the other piers and the main beam (labelled boundary condition B); and (3) the hinged connection was modelled between the pier labelled N5 and the main beam, while the rigid connection was considered between the other piers and the main beam (labelled boundary condition C). The results are plotted in Figures 17–19.

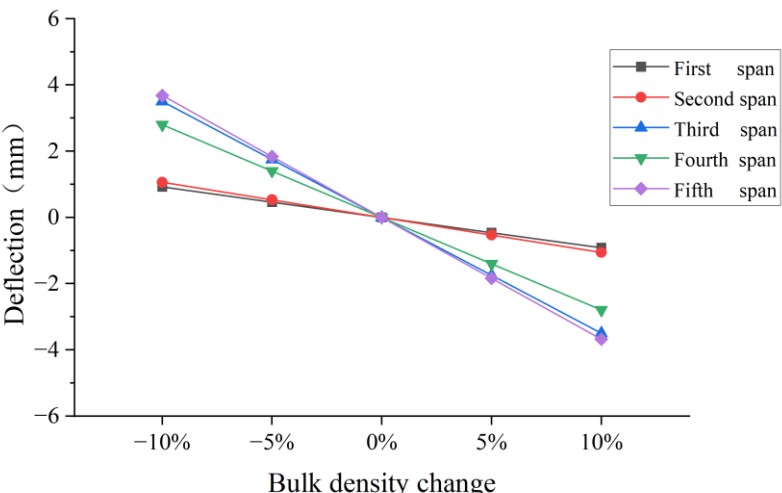

**Figure 17.** Relationship between the bulk density and maximum variation value of deflection per span corresponding to the boundary condition A.

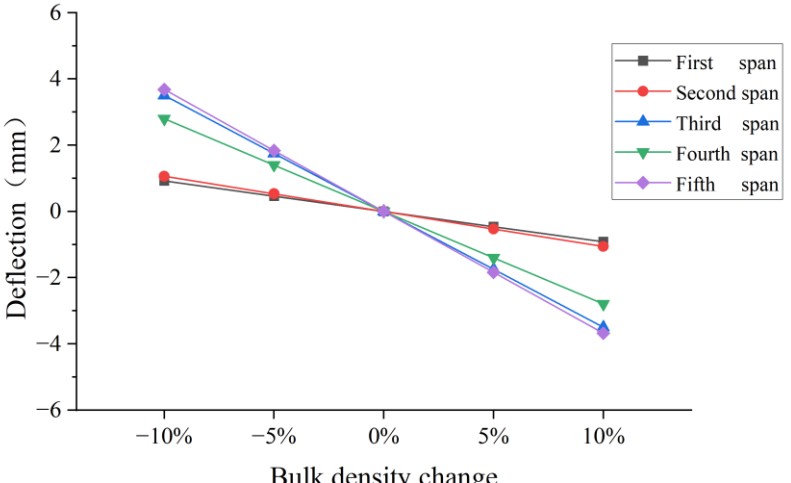

**Figure 18.** Relationship between the bulk density and maximum variation value of deflection per span corresponding to the boundary condition B.

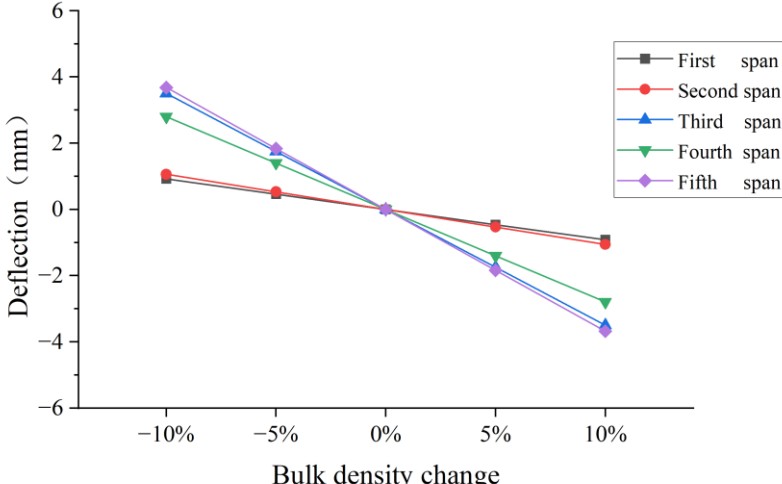

**Figure 19.** Relationship between the bulk density and maximum variation value of deflection per span corresponding to the boundary condition C.

It can be seen from the above figures that under three different boundary conditions, the maximum deflection change in each span still occurs at the mid-span position, and with the change in bulk density, the maximum deflection change (mid-span position) in each span presents a linear relationship, in which the change is the largest at the fifth span, and the deflection changes by 26% when the bulk density changes by 5%. However, the qualitative conclusions are still the same. Therefore, it is believed that the performed sensitivity analyses are valid for various boundary conditions.

## 5. Comparative Analysis of Sensitive Factors

In the actual construction process, some parameters will change due to the influence of various circumstances, which will lead to the deviation from the design stage of the construction, and eventually may lead to a certain deviation between the linear shape of the completed bridge and the theoretical situation. Therefore, the primary problem of construction control is how to adjust the parameters so that the parameters are closer to the actual value to ensure that the stress condition and linear change in the bridge are as close as possible to the theoretical value, and the key variables can be identified. Through sensitivity analysis, the degree of influence of uncertainty on bridge deformation can be assessed and corresponding measures can be taken to reduce the risk caused by uncertainty.

Based on the abovementioned parametric analysis, the effects of five parameters (i.e., the internal prestress, bulk density of the main beam, pipeline friction coefficient, pipeline deviation coefficient, and external stress) on the deflection of the main beam could be analyzed more intuitively, as given in Table 1. As shown in Table 1, the deflection of the bridge is most sensitive to the bulk density, successively the external prestress, internal prestress, pipe friction coefficient, and pipeline deviation coefficient. More attention should be paid to the control of the bulk density and internal and external prestresses of the main beam of the north approach bridge of Nanjing Xinshengwei Yangtze River Bridge during the construction process.

**Table 1.** Statistic results of the parametric analysis.

| Sensitive Factors / Range of Variation | −10% | −5% | 5% | 10% |
|---|---|---|---|---|
| Bulk density | 52% | 26% | −26% | −52% |
| Internal prestress | −10% | −5% | 5% | 10% |
| Pipe friction coefficient | 0.6% | 0.3% | −0.3% | −0.6% |
| Pipeline deviation coefficient | 0.4% | 0.2% | −0.2% | −0.4% |
| External prestress | −12% | −6% | 6% | 12% |

It should be also noted that simplified models are often based on several assumptions and simplifications, which can lead to differences from the actual situation. Therefore, the results of sensitivity analysis may be limited by these assumptions and may not fully accurately reflect the behavior of real systems. Sensitivity analysis requires the selection of parameters to be analyzed, which involves subjective judgment. If the parameters selected are incomplete or inaccurate, the understanding of the system behavior may be incomplete or inaccurate.

## 6. Conclusions

In this paper, an equivalent method for converting the cross-section of corrugated steel webs into those with pure concrete web is proposed based on theoretical derivation. Subsequently, a parametric analysis was conducted to investigate the sensitivity of different structural parameters. The following conclusions can be drawn.

(1) After equivalent conversion, the characteristics of the deflection curve corresponding to the linkage model and solid model are almost the same, and the difference in the maximum deflection is small. The equivalent conversion method can be adopted and the simple linkage model can be used for parametric analysis.

(2)　The deflection of the main beam is most sensitive to the bulk density, followed by external prestress, internal prestress, pipeline friction coefficient, and pipeline deviation coefficient, successively. The bulk density of the main beam and internal and external prestress are all sensitive parameters, while the friction coefficient and pipeline deviation coefficient are non-sensitive parameters.

(3)　The deflection of the main beam varies linearly with the increasing or decreasing value of the bulk density, external prestress, internal prestress, pipe friction coefficient, and pipe deviation coefficient. More attention should be paid to the bulk density and internal and external prestresses during the construction process.

Notably, the present study was carried out based on the static analysis of multi-span bridges. While in practical engineering, bridges are usually subjected to dynamic moving vehicle loads. Therefore, in future studies, the dynamic governing equations (i.e., equations of motion) should be employed to better clarify the structural response of the multi-span prestressed bridges under moving vehicles, i.e., dealing with dynamic response of beam-like elements via various beam theories [29,33–35].

Meanwhile, the present study mainly focuses on the comparison between the bar system element and the solid element model after the completion of the bridge and under the action of dead weight. In future studies, the comparison during the construction process can be further investigated, and the deflection and stress can be compared and analyzed. Additionally, the sensitivity analysis in this paper only focuses a single factor, and multi-factor changes can be investigated in the future. In this case, various new approaches, e.g., deep learning [36–40], artificial neural network [41], etc., enable us consider a wide range of factors in estimating the near-to-exact response of various structures, which warrants further investigations in the future.

**Author Contributions:** M.D. developed the methodology; F.W. was in charge of the formal analysis; Y.W. was responsible for data curation; H.T. and D.Z. mainly worked on visualization. All authors have read and agreed to the published version of the manuscript.

**Funding:** This research received no external funding.

**Institutional Review Board Statement:** Not applicable.

**Informed Consent Statement:** Informed consent was obtained from all subjects involved in this study.

**Data Availability Statement:** The data that support the findings of this study are available from the corresponding author upon reasonable request.

**Conflicts of Interest:** The authors declare no conflict of interest.

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
