# Peer review of "Sensitivity Analysis of Structural Parameters of Unequal-Span Continuous Rigid Frame Bridge with Corrugated Steel Webs"

_applsci, doi:10.3390/app131810024_

Round 1

Reviewer 1 Report

The investigation of the authors into the effect of structural parameters on unequal-span bridges involved creating a finite element model that closely simulated a full-scale bridge, while also considering the rigorous construction process. To streamline the performed analysis, the authors proposed an equivalent method of replacing corrugated steel webs with concrete webs based on theoretical derivation. The carried-out analysis revealed that the material bulk density is the most sensitive parameter to the structural response, followed by the external prestress, internal prestress, pipe friction coefficient, and pipe deviation coefficient. Notably, the bulk density, internal prestress, and external prestress are all highly sensitive parameters, while the pipe friction coefficient and pipe deviation coefficient are non-sensitive. The authors also conducted an extensive analysis on the influences of external prestress on the bridge's main beam after construction. 

The major comments on this work have been itemized as follows: 

  1. It is NOT clear exactly how the plotted deflection as a function of the length of the main beam in Figure 5 based on the “Linkage model” has been extracted. Can the authors explain more or update us with the final relations/equation of deflection in terms of the beam’s length that leads to the plot demonstrated in this figure? Please clarify in some detail. 

2.     No attempts have been made to verify some parts of the obtained results with those of other published works! The authors are highly requested to check some crucial results with those of other researchers. 

3.     What are the main assumptions and limitations as well as the maximum capabilities of the proposed model? Please clarify them in appropriate places within your paper to show the 

4.     The workability and importance of the suggested model have been not deeply explained. In this regard, the authors are highly invited to reveal the limitations of the proposed models.

5.     It would be better to analytically model the deflection of corrugated steel webs via beam models (i.e., Euler-Bernoulli beam theory, Timoshenko beam theory, and Reddy beam theory) in the presence of applied loads, as a reduced and simplified one-dimensional model of the whole three-dimensional one. Can the authors display how it can be performed? In this regard, the following papers deal with vibrations of simply supported and cantilevered beams and their wave propagation problems, from macro to nano, with and without cracks and damages, will be helpful to readers and should be suitably cited and explained within the paper:

(1) https://doi.org/10.1016/j.mechrescom.2014.05.005 

(2) https://doi.org/10.1016/j.jsv.2007.11.010

(3) https://doi.org/10.1016/j.cap.2014.05.018 

(4) https://doi.org/10.1016/j.engfailanal.2013.02.016

(5) https://doi.org/10.1016/j.physe.2014.01.033 

(6) https://doi.org/10.1016/j.engfailanal.2013.02.016 

(7) https://doi.org/10.1007/s11012-014-9957-2 

(8) https://doi.org/10.1109/TIM.2020.3018578 

And the following papers display vibrations of multi-span beam-like structures (i.e., a reduced one-dimensional model for multi-span bridges) using meshless techniques can be also cited:

(1) https://doi.org/10.1007/s10409-010-0365-0 

(2) https://doi.org/10.1115/1.3147165 

(3) https://doi.org/10.1016/j.ymssp.2018.11.056 

(4) https://doi.org/10.1007/s40435-023-01214-5 

6.     What were the specific findings of the sensitivity analysis conducted in this research? Were there any significant correlations found between the investigated parameters and the deflection of the bridge? Were there any limitations to the simple variable method that was adopted for this study? Please clarify all of these quires per simple and short responses in the original text of the paper. 

7.     Are all the performed sensitivity analyses in Figures 7-16 valid for various boundary conditions? Why? Please explain that in more detail. 

8.     What is the significance of the maximum deflection at the cantilever end increasing with increasing external prestress? Why does the maximum variation value of the deflection occur at the mid-span position after changing the external prestress? How does the sensitivity of deflection as a function of variation of the external pressure vary across different spans of the bridge?

9.     The paper suffers from physical interpretations behind the plotted results in Figures 7-16! In fact, the reasons behind the demonstrated trends of the plotted results have been NOT carefully clarified and explained. How the authors can handle this issue? 

10.  What is the next step and what are the possible future works? Please suggest some possible/practical works for the future, encouraging the followers and young investigators to follow up. 

11.  There are some grammatical errors within the manuscript of the paper, requiring high attention of the authors; for example, in the “Abstract” part, the statement “the section of composite beam with corrugated steel web were firstly converted…” should be revised to “the section of composite beam with corrugated steel web was first converted…”. In addition, there are many cases that the article “the” has been missed! Please check the entire manuscript carefully against any grammatical issues for better readability. 

Some of them are given in the above box. In this regard, There are some grammatical errors within the manuscript of the paper, requiring high attention of the authors; for example, in the “Abstract” part, the statement “the section of composite beam with corrugated steel web were firstly converted…” should be revised to “the section of composite beam with corrugated steel web was first converted…”. In addition, there are many cases that the article “the” has been missed! Please check the entire manuscript carefully against any grammatical issues for better readability.

Reviewer 2 Report

In this work, the section of composite beam with corrugated steel web were converted into the section composed of the same material. Moreover, an equivalent method of replacing corrugated steel webs with concrete webs was proposed based on theoretical derivation. As a result, after equivalent replacement, the influences of material bulk density, internal prestress, pipe friction coefficient and pipe deviation coefficient on the main beam at the maximum cantilever stage were analyzed, and the influences of external prestress on the main beam after bridge construction were analyzed. It was concluded that the results show that the most sensitive parameter to structural response is bulk density, subsequently the external prestress, internal prestress, pipe friction coefficient and pipe deviation coefficient. It is a very useful and practical research, and interesting to readers. Moreover, it will have application in the techniques for composite structures. The reviewer found that this manuscript is well organized, and suggest this manuscript can be accepted after minor revisions. My specific comments are as follows,

-This is a very practical study. The author puts forward the possibility of evaluation of the effect of structural parameters of unequal span bridges on the bridge alignment. What is the impact of unequal spans?  

-What is the advantages of the proposed conversion of corrugated steel web into the section composed of the same material? Please give some quantitative analysis by comparison with other methods.

-What is the disadvantages of the proposed model for parameter sensitivity analysis? Anything different with others’ approaches? Please give quantitative analysis and composition.

--The reference part can be improved a lot by adding more research on corrugated structures, for example, DOI: 10.1177/1369433219858451

-Are there any actual data for validation besides simulation?

-Please double check English presentation throughout the context.

Please double check english presentation.

Round 2

Reviewer 1 Report

please see the attached pdf file.

please see the attached pdf file.
